# Neural Injury of the Dopaminergic Pathways in Patients with Middle Cerebral Artery Territory Infarct: A Diffusion Tensor Imaging Study

**DOI:** 10.3390/brainsci13060927

**Published:** 2023-06-08

**Authors:** Jeong Pyo Seo, Heun Jae Ryu

**Affiliations:** 1Department of Physical Therapy, College of Health Sciences, Dankook University, Cheonan 31116, Republic of Korea; raphael0905@hanmail.net; 2Department of Public Health Sciences, Graduate School, Dankook University, Cheonan 31116, Republic of Korea

**Keywords:** diffusion tensor tractography, mesocortical tract, mesolimbic tract, stroke, middle cerebral artery, infarction, ischemic stroke

## Abstract

The mesocortical tract (MCT) and mesolimbic tract (MLT), dopaminergic pathways originating from the ventral tegmental area in the midbrain to the ventral striatum (nucleus accumbens) and prefrontal cortex, play a crucial role in regulating incentive salience. This study aimed to investigate the potential changes in the MCT and MLT pathways following ischemic stroke, such as middle cerebral artery (MCA) infarction. We enrolled thirty-six patients with MCA infarction and forty healthy individuals with no history of psychiatric or neurological disorders. Using diffusion tensor tractography, we examined the injury to the affected and unaffected MCT and MLT pathways in patients with MCA infarction, comparing them to the control group. Our findings revealed a significant difference in the mean values of fractional anisotropy (FA) and tract volume (TV) of the MCT and MLT pathways between the patient and control groups (*p* < 0.05). Specifically, the mean FA of the MCT and MLT showed a decrease of 7.94% and 6.33%, respectively, in the affected side compared to the control group (*p* < 0.05). Similarly, the mean TV of the MCT and MLT showed a decrease of 73.22% and 78.79%, respectively, in the affected side compared to the control group (*p* < 0.05). These changes were significantly different from those of the unaffected MCT, MLT, and control groups (*p* < 0.05). Our study suggests that MCA infarction can cause significant damage to the affected MCT and MLT pathways, potentially contributing to our understanding of the pathophysiology of post-stroke depression.

## 1. Introduction

Strokes are the leading cause of death and main cause of disability worldwide [1,2]. Strokes can be categorized into two main types: hemorrhagic and ischemic, with the latter being more common. A stroke caused by a lack of blood flow to the brain is known as an ischemic stroke [3]. The middle cerebral artery (MCA) region is the most-frequently damaged area of the brain due to ischemic strokes [4]. The MCA supplies blood flow to critical brain areas, such as the caudate nucleus, internal capsule, thalamus, and cerebral cortex [5]. Consequently, those areas are the most-commonly injured areas by ischemic strokes.

Clinical studies have shown that most ischemic strokes cause damage to the subcortical structures, including the basal ganglia [6]. These structures play a crucial role in various functions, including motor control, learning, and emotion regulation. The damage caused by ischemic strokes can therefore lead to a wide range of neurological and cognitive deficits. The mesocortical tract (MCT) and mesolimbic tract (MLT) are dopaminergic reward pathways that originate from the ventral tegmental area (VTA) in the midbrain to the ventral striatum (VS), which is part of the basal ganglia and prefrontal cortex (PFC) [7]. These pathways are integral to our understanding of reward processing, motivation, and emotional regulation in the brain. In addition to regulating incentive salience, the nucleus accumbens (NAc) of the VS is also involved in cognitive control, motivation, and emotional response [7,8]. This region is particularly interesting, as it is implicated in a variety of mental health disorders, including depression, addiction, and schizophrenia. Impairments and alterations in the MCT and MLT pathways are associated with several cognitive and psychiatric disorders, such as depression [9,10]. This suggests that damage to these pathways due to ischemic stroke could potentially lead to, or exacerbate, these disorders. According to previous studies, dopamine signaling in the NAc and post-stroke depression are closely related to the structural abnormalities observed in patients with ischemic stroke [11,12]. This highlights the importance of dopamine signaling in the brain’s response to stroke and its role in post-stroke recovery and mental health.

However, it is uncertain whether ischemic strokes, such as MCA infarction, are associated with depression due to structural abnormalities in the MCT and MLT pathways. This is a critical question that warrants further investigation, as understanding this relationship could have significant implications for the treatment and management of post-stroke depression.

Diffusion tensor imaging (DTI), which provides images based on estimations of water molecule diffusion in microstructures, has permitted complete evaluation of the white matter microstructure in the human brain. DTI-derived diffusion tensor tractography (DTT) allows for the reconstruction and visualization of the MCT and MLT pathways in three dimensions. Although several studies were conducted on the MCT and MLT pathways [9,10,13], no studies have been conducted on the MCT and MLT pathways in patients with MCA infarctions. Therefore, the aim of this study is to investigate the injury that occurs to the MCT and MLT pathways in patients with MCA infarction. Specifically, we are trying to see how MCA affects the structure of these pathways in comparison to those in healthy individuals, using DTT as our primary tool for this investigation.

## 2. Methods

### 2.1. Subjects

Forty-two patients with MCA territory infarction of ischemic stroke (males, 26/42 (61.9%); females, 16/42 (38.1%); mean age, 54.76 years; and range, 22–70 years) and forty normal controls (males, 25/40 (62.5%); females, 15/40 (37.5%); mean age, 55.65 years; and range, 30–79 years), with no previous history of psychiatric or neurological diseases, were enrolled (Table 1). (G-POWER: F tests; ANOVA; Effect size: 0.5, α-err-prob: 0.05, Power [1-β err-prob:0.8] = 42). Six of the forty-two patients had extensive damage and significant degeneration. In conclusion, thirty-six patients were analyzed (males, 20/36 (55.56%); females, 16/36 (44.44%); mean age, 55.89 years; and range, 34–70 years) (Table 1). Accordingly, they were excluded from the analysis. Patients with stroke were consecutively enrolled according to the following inclusion criteria: (1) patients experiencing stroke for the first time; (2) patients aged 20–70 years; (3) a duration between the onset and time of MRI scanning of less than 8 weeks; and (4) infarction affecting the MCA region as confirmed by a neuroradiologist. Normal controls with no previous history of psychiatric, neurological, or physical diseases, and no brain lesions according to the conventional MRI (T1-weighted, T2-weighted, fluid-attenuated inversion recovery, or T2-weighted gradient recall echo images), as confirmed by a neuroradiologist, were recruited. All the participants provided written informed consent prior to their enrollment in the study. The study was conducted in accordance with the Declaration of Helsinki. This study was conducted retrospectively, and approval for the study was obtained from the Institutional Review Board of University Hospital (YUMC-2021-03-014).

### 2.2. Data Acquisition

DTI was performed using a Synergy-L SENSE head coil on a 1.5 T Gyroscan Intera system (Philips, Best, The Netherlands) equipped with a single-shot echo-planar imaging system. For each of the 32 noncollinear diffusion-sensitizing gradients, 67 contiguous slices were acquired parallel to the anterior–posterior commissure line. The imaging parameters were as follows: acquisition matrix = 96 × 96, reconstructed matrix = 192 × 192, field of view = 240 × 240 mm^2^, TR = 10,398 ms, TE = 72 ms, parallel imaging reduction factor (SENSE factor) = 2, EPI factor = 59, *b* = 1000 s/mm^2^, NEX = 1, slice gap = 0, and slice thickness = 2.5 mm.

### 2.3. Fiber Tracking

The Oxford Centre for Functional Magnetic Resonance Imaging of the Brain (FMRIB) Software Library was used to analyze the diffusion-weighted imaging data. For the eddy currents, affine multi-scale two-dimensional registration was used to correct the head-motion effect and image distortion (FSL, www.fmrib.ox.ac.uk/fsl, accessed on 21 April 2023). A probabilistic tractography method based on a multifiber model was used for fiber tracking and applied utilizing tractography routines implemented in FMRIB Diffusion (step length, 0.5 mm; 5000 streamline samples; and curvature threshold, 0.2) [14,15,16,17]. The MCT and MLT pathways were delineated by selecting the fibers that passed through the seed and target regions of interest (ROIs). For each participant, as in the MCT, the seed ROI was placed on the VTA in the midbrain and the target ROI was located in the PFC of the inferior frontal sulcus [18]. For the MLT pathway, the seed ROI was placed on the VTA in the midbrain and the target ROI was located in the NAc of the VS [18]. Out of the 5000 generated samples from each seed voxel, the results for each contact included a visualized threshold and weightings of tract probability for a minimum of one streamline through each analyzed voxel (Figure 1). The analysis involved visualizing the results of the MCT and MLT, selected by passing fibers through the seed and target regions of interest, by applying a threshold level of one streamline per voxel. The widely utilized software tool, MATLABTM (MATLAB R2022a, The Mathworks, Natick, MA, USA), was employed to measure DTT parameters, namely fractional anisotropy (FA) and tract volume (TV). The FA and TV values were ascertained by counting the voxels of the MCT and MLT. This method facilitated a more concentrated analysis of the MCT and MLT, along with the precise calculation of crucial metrics such as the FA and TV.

### 2.4. Statistical Analysis

Statistical Package for the Social Sciences (SPSS) software (version 25.0; SPSS, Chicago, IL, USA) was used for the data analysis. One-way analysis of variance and Bonferroni post-hoc tests were used to determine the statistical significance of differences for each DTI parameter (FA and TV) between the two groups. Statistical significance was set at *p* < 0.05.

## 3. Results

Table 2 shows a comparison between the mean values of the MCT and MLT parameters in the patients in the MCA group (unaffected and affected sides) and the control group. There was a significant difference in the mean FA and TV values in the MCT and MLT between the patient and control groups (*p* < 0.05).

The mean values of FA and TV of the unaffected side of the MCT were significantly lower than those of the unaffected side and control groups (*p* < 0.05). In addition, there was no significant difference in any of the DTI parameters between the unaffected sides of the MCT and control groups (*p* > 0.05). The mean FA value of the affected side of the MLT showed a significant difference compared with those of the unaffected and control groups (*p* < 0.05), and there was no significant difference in the mean FA value between the unaffected side of the MLT and control groups (*p* > 0.05). However, with respect to the mean TV value in the MLT, the TV of the affected side was significantly lower than that of the unaffected side and control groups. Moreover, the TV of the unaffected side was significantly lower than that of the control group (*p* < 0.05).

## 4. Discussion

In the current study, we used DTI to investigate injury of the MCT and MLT pathways in the patients with MCA, which were reconstructed in three dimensions (Figure 1). The mean FA and TV values of the affected side of the MCT and MLT pathways were significantly lower than those of the unaffected side and controls. FA refers to the degree of directionality and integrity of white matter microstructures, such as axons, myelin, and microtubules [19,20], while TV indicates the total number of fibers in a neural pathway [21]. Accordingly, reduced FA and TV values could reflect an injury to the neural tract [22,23].

The MCT and MLT pathways showed microstructural damage as a consequence of infarction in the MCA territory. This damage is indicative of the severe impact of a stroke, particularly an ischemic stroke, on the brain’s structure and function. This damage is not limited to the site of the infarction, but can also affect connected pathways, such as the MCT and MLT pathways in our study. Moreover, the mean values of FA and TV of the affected side of the MCT and MLT pathways were significantly lower than those of the unaffected side of both pathways and controls. The mean TV values of the unaffected side of the MLT pathway were significantly lower than those in the control group. Similarly, significant differences were observed in FA and TV on the affected side compared to the control group and the unaffected side in MCA stroke patients in the dopamine tract, and similar results have been reported [24].

Additionally, it is important to consider the intersection of the central dopamine system and stroke in the context of our findings. Stroke has been shown to trigger an early and massive release of dopamine into the striatum [25,26]. This could possibly exacerbate the microstructural damage in the MCT and MLT pathways observed in our study, as the sudden influx of dopamine might further disrupt the normal functioning of these pathways. Moreover, a decrease in dopamine receptors following stroke has been reported [26,27,28]. This decrease could potentially contribute to the observed lower mean values of FA and TV on the affected side of the MCT and MLT pathways, as the reduced number of dopamine receptors might impair the transmission of signals along these pathways. These findings from the literature further underscore the severe impact of strokes, particularly ischemic strokes, on the brain’s structure and function.

Previous studies reported that corticospinal fibers on the unaffected side were injured due to extensive damage [29]. Therefore, our findings suggest that the mean TV values of the affected side of the MLT pathway might even be lower on the unaffected side due to the wide range of damage. Previous pathological studies have shown neuronal death, gliosis, and axonal degeneration in the basal ganglia and midbrain on the affected side after MCA infarction [30,31,32,33]. The affected side of the MCT and MLT pathways were narrower and more discontinuous than the unaffected side and the control, because the VSA and VS were susceptible to injury during the MCA infarction [31,32,33]. Therefore, these results demonstrate the relationship between MCA infarction and neuropathological alterations in the MCT and MLT pathways. Previous research has attempted to establish that post-stroke depression can be explained by circuits related to the NAc due to volumetric and microstructural alterations [12]. Moreover, subsequent significant alterations in the gray and white matter are likely responsible for the abnormal NAc network dynamics observed in these patients [11]. By examining these alterations, a better understanding of their pathophysiology, including post-stroke depression from infarction, could be grasped [11,12]. Therefore, this finding could be a projection of the results of these prior studies by showing that the connectivity of the structural pathway of the MCT and MLT involves not only neuropathological alterations in the dopaminergic tracts but could also be related to post-stroke depression.

This study had several limitations. First, DTT analysis generated false-positive and false-negative findings, owing to the crossing fibers and partial volume effect, respectively. Second, fiber crossing and complexity may deem the accurate DTT depiction of the underlying fiber architecture difficult. Consequently, the DTT findings may have underestimated the significance of fibers in the neural tract. In addition to the limitations related to DTT analysis and lack of clinical correlation, the study was further constrained by a small sample size and a lack of consideration for variability in age and stroke type among participants, which may have influenced the findings and their generalizability. Future research should overcome these limitations and conduct neuroimaging comparisons using clinical data.

## 5. Conclusions

In summary, the study highlighted the significant differences in the mean values of FA and TV of the MCT and MLT between patients with MCA infarctions of ischemic strokes and control groups. Through the utilization of DTT, these findings demonstrated that the affected side of the MCT and MLT pathways were compromised in the patients with MCA stroke compared with the unaffected side of these patients and control group. This finding provides a fresh perspective on the structural alterations that occur in the brain after an MCA infarction, especially in the MCT and MLT pathways, which have previously received little attention. By analyzing these alterations, a better understanding of the pathophysiology of post-stroke depression following infarction could be reached.

## Figures and Tables

**Figure 1 brainsci-13-00927-f001:**
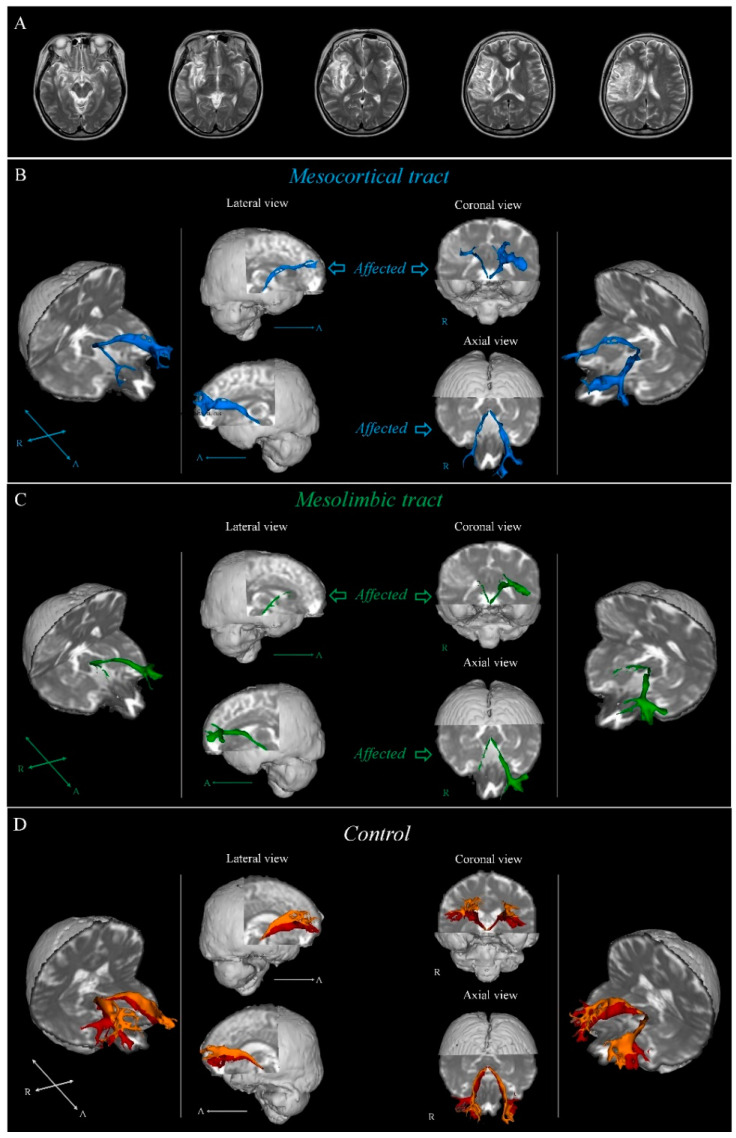
(**A**) T2-weighted brain magnetic resonance images showing a MCA patient (53-year-old female). (**B**) Results for the mesocortical tract of patient on diffusion-tensor tractography; sky blue. (**C**) Results for the mesolimbic tract of patient on diffusion-tensor tractography; green. (**D**) Results for the mesocortical mesolimbic tracts of control on diffusion-tensor tractography; orange—mesocortical tract, red—mesolimbic tract (52-year-old male).

**Table 1 brainsci-13-00927-t001:** Demographic and clinical data of the patients. Values indicate mean (±standard deviation).

	Control	Experimental
Sex (male/female)	25/15	20/16
Mean age (year)	55.65 (12.79)	55.89 (8.80)
Lesion side (right/left)	-	21/15
Onset duration (days)	-	21.4615 (1.7791)

**Table 2 brainsci-13-00927-t002:** Comparison of variations of diffusion tensor trajectory by one-way ANOVA.

		Condition	Mean	F	*p*	Post-Hoc
MCT	FA	Control	0.3881 (0.0490)	5.791	0.004 *	C > U, A
		Unaffected side	0.3883 (0.0314)			U > C, A
		Affected side	0.3573 (0.0569)			C, U > A
	TV	Control	520.81 (357.37)	17.019	<0.001 *	C > U, A
		Unaffected side	411.78 (374.44)			U > C, A
		Affected side	139.50 (152.69)			C, U > A
MLT	FA	Control	0.3822 (0.0411)	4.474	0.013 *	C > U, A
		Unaffected side	0.3867 (0.0469)			U > C, A
		Affected side	0.3580 (0.0529)			C, U > A
	TV	Control	556.74 (349.46)	32.379	<0.001 *	C > U, A
		Unaffected side	289.64 (232.62)			C > U > A
		Affected side	118.03 (123.85)			C, U > A

MCT: mesocortical tract, MLT: mesolimbic tract; FA: fractional anisotropy, TV: tract volume; C: control group; U: unaffected group; A: affected group. Values indicate mean (±standard deviation); * *p* < 0.05.

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
