# Peer review of "Neural Injury of the Dopaminergic Pathways in Patients with Middle Cerebral Artery Territory Infarct: A Diffusion Tensor Imaging Study"

_brainsci, 2023, doi:10.3390/brainsci13060927_

Round 1
Reviewer 1 Report
One of the manuscript's most compelling features is how original the topic is. Despite the above, there is a significant need for improvements to be made to the presentation. In addition, in order to improve the scientific validity, I would suggest to the writers that they include more material drawn from the relevant body of research on the topic. This is because the topic is important, and I believe that it will be of great interest to the audience.
In the Abstract, do not use the chapers' name i.e. "background, methods"
Regarding the Keywords, please do not capitalize each word and do not use abbreviations in the Keywords section, you can do that in the abstract or main text.
Some abbreviations are repeated, i.e. "MCA"
To ensure that the audience has a solid grasp of the material, the "Introduction" needs to be expanded upon and given further details.
In the"Methods" section, it would be helpful to include an "Exclusion criteria" section.
Include more information inside the captions of Figure 1.
It would be useful to include, in the Discussion section, an analysis comparing and contrasting your results with those of other authors' conclusions.
Please include a chapter about the limitations.
It would be helpful if you could create a chapter labeled "Conclusion" and choose parts of the text that include information that the journal's audience would find interesting.
Author Response
Answers to Reviewer 1 Comments
Point 1. In the Abstract, do not use the chapters' name i.e. "background, methods"
Answer 1: I totally agree with the reviewer`s comment. I have made changes by deleting the part you mentioned and modifying it accordingly.
Abstract
Background: The mesocortical tract (MCT) and mesolimbic tract (MLT), which are dopaminergic pathways that originate from the ventral tegmental area in the midbrain to the ventral striatum (nucleus accumbens) and prefrontal cortex, regulate incentive salience. However, it is unclear whether the effects of ischemic stroke, such as middle cerebral artery (MCA) infarction, are associated with changes in the MCT and MLT pathways.
Methods: Thirty-six patients with MCA infarction and 40 healthy individuals with no history of psychiatric or neurological disorders were enrolled in this study. Diffusion tensor tractography was used to investigate injury to the affected and unaffected MCT and MLT pathways in the patients with MCA infarction compared with those in the normal human brain.
Results: There was a significant difference in the mean values of fractional anisotropy (FA) and tract volume (TV) of the MCT and MLT pathways between the patient and control groups (p<0.05). Post hoc analysis revealed that the mean FA and TV from the affected MCT and MLT were significantly different from those of the unaffected MCT, MLT, and control groups (p<0.05).
Conclusion: MCA infarction can damage the affected MCT and MLT pathways. Our study findings could provide a better understanding of the pathophysiology of post-stroke depression.
Point 2. Regarding the Keywords, please do not capitalize each word and do not use abbreviations in the Keywords section, you can do that in the abstract or main text.
Answer 2: I totally agree with the reviewer`s comment. I have made changes by deleting the part you mentioned and modifying it accordingly.
Keywords: diffusion tensor tractography; mesocortical tract (MCT); mesolimbic tract (MLT); stroke; middle cerebral artery (MCA); infarct; ischemic stroke
Point 3. To ensure that the audience has a solid grasp of the material, the "Introduction" needs to be expanded upon and given further details.
Answer 3: I totally agree with the reviewer`s comment. I have added and modified the part you mentioned.
Introduction
Stroke is the leading cause of death and main cause of disability worldwide [1,2]. Stroke can be categorized into two main types: hemorrhagic and ischemic, with the latter being more common. Stroke caused by a lack of blood flow to the brain is known as ischemic stroke [3].
…
Clinical studies have shown that most ischemic strokes cause damage to the subcortical structures including the basal ganglia [6]. These structures play a crucial role in various functions including motor control, learning, and emotion regulation. The damage caused by ischemic strokes can therefore lead to a wide range of neurological and cognitive deficits. The mesocortical tract (MCT) and mesolimbic tract (MLT) are dopaminergic reward pathways that originate from the ventral tegmental area (VTA) in the midbrain to the ventral striatum (VS), which is part of the basal ganglia and prefrontal cortex (PFC) [7]. These pathways are integral to our understanding of reward processing, motivation, and emotional regulation in the brain. In addition to regulating incentive salience, the nucleus accumbens (NAc) of the VS is also involved in cognitive control, motivation, and emotional response [7,8]. This region is particularly interesting as it is implicated in a variety of mental health disorders, including depression, addiction, and schizophrenia. Impairments and alterations in the MCT and MLT pathways are associated with several cognitive and psychiatric disorders such as depression [9,10]. This suggests that damage to these pathways due to ischemic stroke could potentially lead to or exacerbate these disorders. According to previous studies, dopamine signaling in the NAc and post-stroke depression are closely related to the structural abnormalities observed in the patients with ischemic stroke [11,12]. This highlights the importance of dopamine signaling in the brain's response to stroke and its role in post-stroke recovery and mental health.
However, it is uncertain whether ischemic strokes, such as MCA infarction, are associated with depression due to structural abnormalities in the MCT and MLT pathways. This is a critical question that warrants further investigation, as understanding this relationship could have significant implications for the treatment and management of post-stroke depression.
Point 4. In the"Methods" section, it would be helpful to include an "Exclusion criteria" section.
Answer 4: I totally agree with the reviewer`s comment. I have added and modified the part you mentioned.
Methods
subjects
Forty-two patients with MCA territory infarction of ischemic stroke (males, 26/42 (61.9%); females, 16/42 (38.1%); mean age, 54.76 years; and range, 22–70 years) and 40 normal controls (males, 25/40 (62.5%); females, 15/40 (37.5%); mean age, 55.65 years; and range, 30–79 years), with no previous history of psychiatric or neurological diseases, were enrolled. (G-POWER: F tests; ANOVA; Effect size: 0.5, α-err-prob: 0.05, Power [1-β err-prob:0.8] = 42). Six of the 42 patients had extensive damage and significant degeneration. In conclusion, thirty-six patients were analyzed (males, 20/36 (55.56%); females, 16/36 (44.44%); mean age, 55.23 years; and range, 34–70 years).
- Include more information inside the captions of Figure 1.
Answer 5: I totally agree with the reviewer`s comment. I have added and modified the part you mentioned.
Figure caption
Figure 1. (A) T2-weighted brain magnetic resonance images showing a MCA patient (53-year-old female). (B) Results for the mesocortical tract of patient on diffusion-tensor tractography; sky blue. (C) Results for the mesolimbic tract of patient on diffusion-tensor tractography; green. (D) Results for the mesocortical mesolimbic tracts of control on diffusion-tensor tractography; orange – mesocortical tract, red – mesolimbic tract (52-year-old male).
Point 6. It would be useful to include, in the Discussion section, an analysis comparing and contrasting your results with those of other authors' conclusions.
Answer 6: I totally agree with the reviewer`s comment. I have added and modified the part you mentioned.
Discussion
MCT and MLT pathways were significantly lower than those of the unaffected side of both pathways and controls. The mean TV values of the unaffected side of the MLT pathway were significantly lower than those in the control group. Similarly, significant differences were observed in FA and TV on the affected side compared to the control group and the unaffected side in MCA stroke patients in the dopamine tract, and similar results have been reported [24].
- Seo, J. P.; Koo, D. K. Degeneration of nigrostriatal pathway in patients with middle cerebral infarct: A diffusion tensor imaging study. 2023, 102,14.
Point 7. Please include a chapter about the limitations.
Answer 7: I totally agree with the reviewer`s comment. I have added and modified the part you mentioned.
Limitation
This study had several limitations. First, DTT analysis generated false-positive and false-negative findings owing to the crossing fibers and partial volume effect, respectively. Second, fiber crossing and complexity may deem the accurate DTT depiction of the underlying fiber architecture difficult. Consequently, the DTT findings may have underestimated the significance of fibers in the neural tract. Third, clinical data cannot be correlated with the MCT and MLT pathways owing to the lack of clinical evidence. Future research should overcome these limitations and conduct neuroimaging comparisons using clinical data.

Reviewer 2 Report
This topic can be very interesting, but paper needs some revisions:
- Lines 54-56: "Therefore, this study aimed at investigating the injury that occurs to the MCT and MLT pathways in the patients with MCA infarction compared with the corresponding pathways in the healthy human brain using DTT" What kind of investigation do the authors want to provide? what is the purpose of this paper ?
- Table 2 is very difficult to understand. Can the authors revise it or provide a new version? Can the authors add more details in the table legend?
- Lines 136-137: "The MCT and MLT pathways showed microstructural damage as a consequence of infarction in the MCA territory" Stroke must me discuss more. Some important papers should be considered, look at doi: 10.3390/neurolint14020032 -- doi: 10.1016/j.pediatrneurol.2023.03.005
- Authors must add a "limitations of the study" section at the end of the discussion. Small sample, different age and kind of stroke. Discuss.
- Conclusion is too small. What does this paper add new to the current literature? What do authors propose new?
Minor editing of English language required
Author Response
Answers to Reviewer 2 Comments
Point 1. Lines 54-56: "Therefore, this study aimed at investigating the injury that occurs to the MCT and MLT pathways in the patients with MCA infarction compared with the corresponding pathways in the healthy human brain using DTT" What kind of investigation do the authors want to provide? what is the purpose of this paper?
Answer 1: I totally agree with the reviewer`s comment. I have added and modified the part you mentioned.
Diffusion tensor imaging (DTI), which provides images based on estimations of water molecule diffusion in microstructures, has permitted complete evaluation of the white matter microstructure in the human brain. DTI-derived diffusion tensor tractography (DTT) allows for the reconstruction and visualization of the MCT and MLT pathways in three dimensions. Although several studies were conducted on the MCT and MLT pathways [9,10,13], no studies have been conducted on the MCT and MLT pathways in the patients with MCA infarctions. Therefore, the aim of this study is to investigate the injury that occurs to the MCT and MLT pathways in patients with MCA infarction. Specifically, we're trying to see how MCA affects the structure of these pathways in comparison to those in healthy individuals, using DTT as our primary tool for this investigation.
Point 2. Table 2 is very difficult to understand. Can the authors revise it or provide a new version? Can the authors add more details in the table legend?
Answer 2: I totally agree with the reviewer`s comment. I have added and modified the part you mentioned.
Table.2 Comparison of Variations of Diffusion Tensor Trajectory by One-way ANOVA
|
|
Condition |
Mean |
F |
p |
Post-hoc |
|
|
MCT |
FA |
Control |
0.3881 (0.0490) |
5.791 |
0.004* |
C>A |
|
Unaffected side |
0.3883 (0.0314) |
U>A |
||||
|
Affected side |
0.3573 (0.0569) |
C,U>A |
||||
|
TV |
Control |
520.81 (357.37) |
17.019 |
<0.001* |
C>A |
|
|
Unaffected side |
411.78 (374.44) |
U>A |
||||
|
Affected side |
139.50 (152.69) |
C,U>A |
||||
|
MLT |
FA |
Control |
0.3822 (0.0411) |
4.474 |
0.013* |
C> A |
|
Unaffected side |
0.3867 (0.0469) |
U>A |
||||
|
Affected side |
0.3580 (0.0529) |
C,U>A |
||||
|
TV |
Control |
556.74 (349.46) |
32.379 |
<0.001* |
C>U,A |
|
|
Unaffected side |
289.64 (232.62) |
C>U>A |
||||
|
Affected side |
118.03 (123.85) |
C,U>A |
||||
MCT: mesocortical tract, MLT: mesolimbic tract; FA: fractional anisotropy, TV: tract volume; C: control group; U: unaffected group; A: affected group. Values indicate mean (±standard deviation); *p<0.05
Point 3. Lines 136-137: "The MCT and MLT pathways showed microstructural damage as a consequence of infarction in the MCA territory" Stroke must me discuss more. Some important papers should be considered, look at doi: 10.3390/neurolint14020032 -- doi: 10.1016/j.pediatrneurol.2023.03.005
Answer 3: I totally agree with the reviewer`s comment. I have added and modified the part you mentioned.
The MCT and MLT pathways showed microstructural damage as a consequence of infarction in the MCA territory. This damage is indicative of the severe impact of stroke, particularly ischemic stroke, on the brain's structure and function. This damage is not limited to the site of the infarction but can also affect connected pathways, such as the MCT and MLT pathways in our study.
Point 4. Authors must add a "limitations of the study" section at the end of the discussion. Small sample, different age and kind of stroke. Discuss.
Answer 4: I totally agree with the reviewer`s comment. I have added and modified the part you mentioned.
Limitation
This study had several limitations. First, DTT analysis generated false-positive and false-negative findings owing to the crossing fibers and partial volume effect, respectively. Second, fiber crossing and complexity may deem the accurate DTT depiction of the underlying fiber architecture difficult. Consequently, the DTT findings may have underestimated the significance of fibers in the neural tract. In addition to the limitations related to DTT analysis and lack of clinical correlation, the study was further constrained by a small sample size and lack of consideration for variability in age and stroke type among participants, which may have influenced the findings and their generalizability. Future research should overcome these limitations and conduct neuroimaging comparisons using clinical data.
Point 5. Conclusion is too small. What does this paper add new to the current literature? What do authors propose new?
Answer 5: I totally agree with the reviewer`s comment. I have added and modified the part you mentioned.
Conclusion
In summary, the study highlighted the significant differences in the mean values of FA and TV of the MCT and MLT between the patient with MCA infarction of ischemic stroke and control groups. Through the utilization of DTT, these findings demonstrated that the affected side of the MCT and MLT pathways were compromised in the patients with MCA stroke compared with the unaffected side of these patients and control group. This finding provides a fresh perspective on the structural alterations that occur in the brain after an MCA infarction, especially in the MCT and MLT pathways, which have previously received little attention. By analyzing these alterations, a better understanding of the pathophysiology of post-stroke depression following infarction could be reached.

Reviewer 3 Report
The role is interesting, and it is well developed by having comparison between people with pathology and without pathology.
The statistics are correct, and the subsequent conclusions are correct.
In the introduction, I suggest this quote, to relate stroke with reaction time: PMID: 31909910
One suggestion is to name the control group as people without stroke, but not as normal: "compared with those in the normal human brain"
I suggest putting the abbreviations somewhere else, instead of at the end.
The post hoc table, I suggest improving it, for aesthetics.
Author Response
Answers to Reviewer 3 Comments
Point 1. In the introduction, I suggest this quote, to relate stroke with reaction time: PMID: 31909910
Answer 1: I totally agree with the reviewer`s comment. I have modified the part you mentioned.
Introduction
Stroke is the leading cause of death and main cause of disability worldwide [1,2]. Stroke can be categorized into two main types: hemorrhagic and ischemic, with the latter being more common. Stroke caused by a lack of blood flow to the brain is known as ischemic stroke [3]. The middle cerebral artery (MCA) region is the most frequently damaged area of the brain due to ischemic stroke [4]. The MCA supplies blood flow to critical brain areas such as the caudate nucleus, internal capsule, thalamus, and cerebral cortex [5]. Consequently, those areas are the most commonly injured areas by ischemic strokes.
Clinical studies have shown that most ischemic strokes cause damage to the subcortical structures including the basal ganglia [6]. These structures play a crucial role in various functions including motor control, learning, and emotion regulation. The damage caused by ischemic strokes can therefore lead to a wide range of neurological and cognitive deficits. The mesocortical tract (MCT) and mesolimbic tract (MLT) are dopaminergic reward pathways that originate from the ventral tegmental area (VTA) in the midbrain to the ventral striatum (VS), which is part of the basal ganglia and prefrontal cortex (PFC) [7]. These pathways are integral to our understanding of reward processing, motivation, and emotional regulation in the brain. In addition to regulating incentive salience, the nucleus accumbens (NAc) of the VS is also involved in cognitive control, motivation, and emotional response [7,8]. This region is particularly interesting as it is implicated in a variety of mental health disorders, including depression, addiction, and schizophrenia. Impairments and alterations in the MCT and MLT pathways are associated with several cognitive and psychiatric disorders such as depression [9,10]. This suggests that damage to these pathways due to ischemic stroke could potentially lead to or exacerbate these disorders. According to previous studies, dopamine signaling in the NAc and post-stroke depression are closely related to the structural abnormalities observed in the patients with ischemic stroke [11,12]. This highlights the importance of dopamine signaling in the brain's response to stroke and its role in post-stroke recovery and mental health.
However, it is uncertain whether ischemic strokes, such as MCA infarction, are associated with depression due to structural abnormalities in the MCT and MLT pathways. This is a critical question that warrants further investigation, as understanding this relationship could have significant implications for the treatment and management of post-stroke depression.
Point 2. One suggestion is to name the control group as people without stroke, but not as normal: "compared with those in the normal human brain"
Answer 2: I totally agree with the reviewer`s comment. I have modified the part you mentioned.
Diffusion tensor tractography was used to investigate injury to the affected and unaffected MCT and MLT pathways in the patients with MCA infarction compared with those in the control group as people without stroke.
Point 3. I suggest putting the abbreviations somewhere else, instead of at the end.
Answer 3: Thank you for your comment. I have modified the part you mentioned.
Point 4. Authors must add a "limitations of the study" section at the end of the discussion. Small sample, different age and kind of stroke. Discuss.
Answer 4: I totally agree with the reviewer`s comment. I have added and modified the part you mentioned.
Limitation
This study had several limitations. First, DTT analysis generated false-positive and false-negative findings owing to the crossing fibers and partial volume effect, respectively. Second, fiber crossing and complexity may deem the accurate DTT depiction of the underlying fiber architecture difficult. Consequently, the DTT findings may have underestimated the significance of fibers in the neural tract. In addition to the limitations related to DTT analysis and lack of clinical correlation, the study was further constrained by a small sample size and lack of consideration for variability in age and stroke type among participants, which may have influenced the findings and their generalizability. Future research should overcome these limitations and conduct neuroimaging comparisons using clinical data.
Point 5. The post hoc table, I suggest improving it, for aesthetics.
Answer 5: I totally agree with the reviewer`s comment. I have added and modified the part you mentioned.
Table.2 Comparison of Variations of Diffusion Tensor Trajectory by One-way ANOVA
|
|
Condition |
Mean |
F |
p |
Post-hoc |
|
|
MCT |
FA |
Control |
0.3881 (0.0490) |
5.791 |
0.004* |
C>A |
|
Unaffected side |
0.3883 (0.0314) |
U>A |
||||
|
Affected side |
0.3573 (0.0569) |
C,U>A |
||||
|
TV |
Control |
520.81 (357.37) |
17.019 |
<0.001* |
C>A |
|
|
Unaffected side |
411.78 (374.44) |
U>A |
||||
|
Affected side |
139.50 (152.69) |
C,U>A |
||||
|
MLT |
FA |
Control |
0.3822 (0.0411) |
4.474 |
0.013* |
C> A |
|
Unaffected side |
0.3867 (0.0469) |
U>A |
||||
|
Affected side |
0.3580 (0.0529) |
C,U>A |
||||
|
TV |
Control |
556.74 (349.46) |
32.379 |
<0.001* |
C>U,A |
|
|
Unaffected side |
289.64 (232.62) |
C>U>A |
||||
|
Affected side |
118.03 (123.85) |
C,U>A |
||||
MCT: mesocortical tract, MLT: mesolimbic tract; FA: fractional anisotropy, TV: tract volume; C: control group; U: unaffected group; A: affected group. Values indicate mean (±standard deviation); *p<0.05

Reviewer 4 Report
1. Methodology
a. From where came the individuals?
b. How were these individuals selected?
c. Please, provide a table with all the baseline characteristics of both groups.
d. Were all types of stroke included? How were the individuals assessed regarding the etiology? Could the authors explain why the etiology would not affect the results?
2. Statistics
a. How was the power of the study calculated? Why was selected this number of individuals?
b. Provide a full description of the statistical software. Avoid abbreviations.
c. How was data distributed?
d. How did the authors assess confounding variables?
e. Why was only one-way ANOVA performed?
f. The authors should describe how were the tractography parameters assessed.
3. Discussion
a. A specific paragraph for the limitations should be provided.
b. It is advised to include in the limitations: a lower number of participants, a selected population, uncontrolled confounding variables (etiology)
c. A specific paragraph for the conclusion should be done.
4. Ethics
a. IRB number should be provided.
5. Grammar and misspellings
a. Revise L. 83
6. Other
a. How does this manuscript’s aim differ from others in the literature? It is well known that specific lesions in the cortex can lead to specific damage in different tracts.
7. References
a. The authors should revise the references. It is advised to include the journal of publication.
5. Grammar and misspellings
a. Revise L. 83
Author Response
Answers to Reviewer 4 Comments
Point 1. Methodology
- From where came the individuals?
- How were these individuals selected?
- Please, provide a table with all the baseline characteristics of both groups.
- Were all types of stroke included? How were the individuals assessed regarding the etiology?
Answer 1: Thank you for your comment. I have added and modified the part you mentioned.
|
|
Control |
Experimental |
|
Sex (male/female) |
25/15 |
20/16 |
|
Mean age (year) |
55.65 (12.79) |
55.89 (8.80) |
|
Lesion side (right/left) |
- |
21/15 |
|
Onset duration (days) |
- |
21.4615 (1.7791) |
a, b.
Methods
All the participants provided written informed consent prior to their enrollment in the study. The study was conducted in accordance with the Declaration of Helsinki. This study was conducted retrospectively, and approval for the study was obtained from the Institutional Review Board of University Hospital (YUMC-2021-03-014).
- Table.1 Demographic and clinical data of the patients
Values indicate mean (±standard deviation)
- Forty-two patients with MCA territory infarction of ischemic stroke (males, 26/42 (61.9%); females, 16/42 (38.1%); mean age, 54.76 years; and range, 22–70 years) and 40 normal controls (males, 25/40 (62.5%); females, 15/40 (37.5%); mean age, 55.65 years; and range, 30–79 years), with no previous history of psychiatric or neurological diseases, were enrolled.
Point 2. Statistics
- How was the power of the study calculated? Why was selected this number of individuals?
- Provide a full description of the statistical software. Avoid abbreviations.
- How was data distributed?
- How did the authors assess confounding variables?
- Why was only one-way ANOVA performed?
- The authors should describe how were the tractography parameters assessed.
Answer 2: Thank you for your comment. I have modified the part you mentioned.
- Forty-two patients with MCA territory infarction of ischemic stroke (males, 26/42 (61.9%); females, 16/42 (38.1%); mean age, 54.76 years; and range, 22–70 years) and 40 normal controls (males, 25/40 (62.5%); females, 15/40 (37.5%); mean age, 55.65 years; and range, 30–79 years), with no previous history of psychiatric or neurological diseases, were enrolled. (G-POWER: F tests; ANOVA; Effect size: 0.5, α-err-prob: 0.05, Power [1-β err-prob:0.8] = 42).
- Statistical Package for the Social Sciences (SPSS) software (version 25.0; SPSS, Chicago, IL, USA) was used for the data analysis.
- This study was conducted retrospectively.
d, f. The analysis involved visualizing the results of the MCT and MLT, selected by passing fibers through the seed and target regions of interest, by applying a threshold level of one streamline per voxel. The widely utilized software tool, MATLABTM (Matlab R2022a, The Mathworks, Natick, MA, USA), was employed to measure DTT parameters, namely Fractional Anisotropy (FA) and Tract Volume (TV). The FA and TV values were ascertained by counting the voxels of the MCT and MLT. This method facilitated a more concentrated analysis of the MCT and MLT, along with the precise calculation of crucial metrics such as the FA and TV.
Multiple Group Comparison: In this study, we are comparing the affected and unaffected sides of patients with MCA stroke, as well as a control group. When comparing three or more groups, One-way ANOVA is suitable.
Continuous Dependent Variables: FA (Fractional Anisotropy) and TV (Tract Volume) are continuous dependent variables. One-way ANOVA is suitable for comparing the means of such variables.
Independent Groups: The affected and unaffected sides of patients with MCA stroke, and the control group, are independent groups. One-way ANOVA is used to compare the mean differences between independent groups.
Therefore, in this study, it is appropriate to use One-way ANOVA to compare the differences in FA and TV among the three groups.
And, Based on previous studies, a one-way ANOVA was applied
Seo, J. P.; Koo, D. K. Degeneration of nigrostriatal pathway in patients with middle cerebral infarct: A diffusion tensor imaging study. Medicine 2023, 102,14
Point 3. Discussion
- A specific paragraph for the limitations should be provided.
- It is advised to include in the limitations: a lower number of participants, a selected population, uncontrolled confounding variables (etiology)
- A specific paragraph for the conclusion should be done.
Answer 3 Thank you for your comment. I have added and modified the part you mentioned.
Conclusion
In summary, the study highlighted the significant differences in the mean values of FA and TV of the MCT and MLT between the patient with MCA infarction of ischemic stroke and control groups. Through the utilization of DTT, these findings demonstrated that the affected side of the MCT and MLT pathways were compromised in the patients with MCA stroke compared with the unaffected side of these patients and control group. This finding provides a fresh perspective on the structural alterations that occur in the brain after an MCA infarction, especially in the MCT and MLT pathways, which have previously received little attention. By analyzing these alterations, a better understanding of the pathophysiology of post-stroke depression following infarction could be reached.
Limitation
This study had several limitations. First, DTT analysis generated false-positive and false-negative findings owing to the crossing fibers and partial volume effect, respectively. Second, fiber crossing and complexity may deem the accurate DTT depiction of the underlying fiber architecture difficult. Consequently, the DTT findings may have underestimated the significance of fibers in the neural tract. In addition to the limitations related to DTT analysis and lack of clinical correlation, the study was further constrained by a small sample size and lack of consideration for variability in age and stroke type among participants, which may have influenced the findings and their generalizability. Future research should overcome these limitations and conduct neuroimaging comparisons using clinical data.
Point 4. Ethics
- IRB number should be provided.
Answer 4: I totally agree with the reviewer`s comment. I have added and modified the part you mentioned.
All the participants provided written informed consent prior to their enrollment in the study. The study was conducted in accordance with the Declaration of Helsinki. This study was conducted retrospectively, and approval for the study was obtained from the Institutional Review Board of University Hospital (YUMC-2021-03-014).
Point 5. Grammar and misspellings
- Revise L. 83
Answer 5: Thank you for your comment. I have modified the part you mentioned.
Forty-two patients with MCA territory infarction (males, 26/42 (61.9%); females, 16/42 (38.1%); mean age, 54.76 years; and range, 22–70 years) and 40 normal controls (males, 25/40 (62.5%); females, 15/40 (37.5%); mean age, 55.65 years; and range, 30–79 years), with no previous history of psychiatric or neurological diseases, were enrolled.
Point 6. Other
- How does this manuscript’s aim differ from others in the literature? It is well known that specific lesions in the cortex can lead to specific damage in different tracts.
Answer 6: I totally agree with the reviewer`s comment. I have added and modified the part you mentioned.
Therefore, the aim of this study is to investigate the injury that occurs to the MCT and MLT pathways in patients with MCA infarction. Specifically, we're trying to see how MCA affects the structure of these pathways in comparison to those in healthy individuals, using DTT as our primary tool for this investigation.
Point 7. References
- The authors should revise the references. It is advised to include the journal of publication
Answer 7: I totally agree with the reviewer`s comment. I have added and modified the part you mentioned.
References
- Lozano, R.; Naghavi, M.; Foreman, K.; Lim, S.; Shibuya, K.; Aboyans, V.; Abraham, J.; Adair, T.; Aggarwal, R.; Ahn, S. Global and regional mortality from 235 causes of death for 20 age groups in 1990 and 2010: a systematic analysis for the Global Burden of Disease Study 2010. Lancet. 2012, 380, 2095-2128.
- Murphy, S.L.; Kochanek, K.D.; Xu, J.; Arias, E. Mortality in the United States, 2020. 2021.
- Murphy, T.H.; Corbett, D. Plasticity during stroke recovery: from synapse to behaviour. Nat. Rev. Neurosci. 2009, 10, 861-872.
- Nogles, T.E.; Galuska, M.A. Middle Cerebral Artery Stroke. In StatPearls; StatPearls Publishing: Treasure Island, FL, USA, 2022..
- Dalley, A.F.; Agur, A. Moore’s Clinically Oriented Anatomy, 7th ed.; Lippincott Williams and Wilkins: Philadelphia, PA, USA, 2021
- Corbetta, M.; Ramsey, L.; Callejas, A.; Baldassarre, A.; Hacker, C.D.; Siegel, J.S.; Astafiev, S.V.; Rengachary, J.; Zinn, K.; Lang, C.; et al. Common behavioral clusters and subcortical anatomy in stroke. Neuron. 2015, 85, 927-941.
- Ikemoto, S. Dopamine reward circuitry: two projection systems from the ventral midbrain to the nucleus accumbens–olfactory tubercle complex. Brain Res. Rev. 2007, 56, 27-78.
- Nestler, E.J.; Hyman, S.E.; Holtzman, D.M.; Malenka, R.C. Molecular Neuropharmacology: A Foundation for Clinical Neuroscience, 3rd ed.; McGraw-Hill Medical: New York, NY, USA, 2015.
- Meyer-Lindenberg, A.; Miletich, R.S.; Kohn, P.D.; Esposito, G.; Carson, R.E.; Quarantelli, M.; Weinberger, D.R.; Berman, K.F. Reduced prefrontal activity predicts exaggerated striatal dopaminergic function in schizophrenia. Nat. Neurosci. 2002, 5, 267-271.
- Nestler, E.J.; Carlezon Jr, W.A. The mesolimbic dopamine reward circuit in depression. Biol. Psychiatry. 2006, 59, 1151-1159.
- Kronenberg, G.; Balkaya, M.; Prinz, V.; Gertz, K.; Ji, S.; Kirste, I.; Heuser, I.; Kampmann, B.; Hellmann-Regen, J.; Gass, P; et al. Exofocal dopaminergic degeneration as antidepressant target in mouse model of poststroke depression. Biol. Psychiatry 2012, 72, 273-281.
- Oestreich, L.K.; Wright, P.; O’Sullivan, M.J. Hyperconnectivity and altered interactions of a nucleus accumbens network in post-stroke depression. Brain commun. 2022, 4, fcac281.
- Supekar, K.; Kochalka, J.; Schaer, M.; Wakeman, H.; Qin, S.; Padmanabhan, A.; Menon, V. Deficits in mesolimbic reward pathway underlie social interaction impairments in children with autism. Brain 2018, 141, 2795-2805.
- Behrens, T.E.; Berg, H.J.; Jbabdi, S.; Rushworth, M.F.; Woolrich, M.W. Probabilistic diffusion tractography with multiple fibre orientations: What can we gain? Neuroimage 2007, 34, 144-155.
- Behrens, T.E.; Johansen-Berg, H.; Woolrich, M.; Smith, S.; Wheeler-Kingshott, C.; Boulby, P.; Barker, G.; Sillery, E.; Sheehan, K.; Ciccarelli, O; et al. Non-invasive mapping of connections between human thalamus and cortex using diffusion imaging. Nat. Neurosci. 2003, 6, 750-757.
- Kunimatsu, A.; Aoki, S.; Masutani, Y.; Abe, O.; Hayashi, N.; Mori, H.; Masumoto, T.; Ohtomo, K. The optimal trackability threshold of fractional anisotropy for diffusion tensor tractography of the corticospinal tract. Magn. Reson. Med. Sci. 2004, 3, 11-17.
- Smith, S.M.; Jenkinson, M.; Woolrich, M.W.; Beckmann, C.F.; Behrens, T.E.; Johansen-Berg, H.; Bannister, P.R.; De Luca, M.; Drobnjak, I.; Flitney, D.E; et al. Advances in functional and structural MR image analysis and implementation as FSL. Neuroimage 2004, 23, S208-S219.
- Nakamura-Palacios, E.M.; Lopes, I.B.C.; Souza, R.A.; Klauss, J.; Batista, E.K.; Conti, C.L.; Moscon, J.A.; de Souza, R.S.M. Ventral medial prefrontal cortex (vmPFC) as a target of the dorsolateral prefrontal modulation by transcranial direct current stimulation (tDCS) in drug addiction. J. Neural Transm. 2016, 123, 1179-1194.
- Assaf, Y.; Pasternak, O. Diffusion tensor imaging (DTI)-based white matter mapping in brain research: a review. J. Mol. Neurosci. 2008, 34, 51-61.
- Neil, J.J. Diffusion imaging concepts for clinicians. 2008, 27, 1-7.
- Pagani, E.; Agosta, F.; Rocca, M.A.; Caputo, D.; Filippi, M. Voxel-based analysis derived from fractional anisotropy images of white matter volume changes with aging. Neuroimage 2008, 41, 657-667.
- Bennett, I.J.; Madden, D.J.; Vaidya, C.J.; Howard, D.V.; Howard Jr, J.H. Age‐related differences in multiple measures of white matter integrity: A diffusion tensor imaging study of healthy aging. Hum. Brain Mapp. 2010, 31, 378-390.
- Inglese, M.; Ge, Y. Quantitative MRI: hidden age-related changes in brain tissue. Top Magn. Reson. Imaging 2004, 15, 355-363.
- Seo, J. P.; Koo, D. K. Degeneration of nigrostriatal pathway in patients with middle cerebral infarct: A diffusion tensor imaging study. Medicine 2023, 102,14.
- Kwak, S.Y.; Yeo, S.S.; Choi, B.Y.; Chang, C.H.; Jang, S.H. Corticospinal tract change in the unaffected hemisphere at the early stage of intracerebral hemorrhage: a diffusion tensor tractography study. Eur. Neurol. 2010, 63, 149-153.
- Block, F.; Dihne, M.; Loos, M. Inflammation in areas of remote changes following focal brain lesion. Prog. Neurobiol. 2005, 75, 342-365.
- Buss, A.; Pech, K.; Merkler, D.; Kakulas, B.A.; Martin, D.; Schoenen, J.; Noth, J.; Schwab, M.E.; Brook, G.A. Sequential loss of myelin proteins during Wallerian degeneration in the human spinal cord. Brain 2005, 128, 356-364.
- Forno, L.S. Reaction of the substantia nigra to massive basal ganglia infarction. Acta Neuropathol. 1983, 62, 96-102.
- Ogawa, T.; Yoshida, Y.; Okudera, T.; Noguchi, K.; Kado, H.; Uemura, K. Secondary thalamic degeneration after cerebral infarction in the middle cerebral artery distribution: evaluation with MR imaging. Radiology 1997, 204, 255-262.

Round 2
Reviewer 2 Report
good
Author Response
Dear Reviewer,
Thank you for the positive response.
Best regards

Reviewer 4 Report
1) Abstract. Could the authors provide more data in the abstract? The data should be in numbers and percentages.
2) Please limitations should be described in the last paragraph of the discussion.
3) It is advised to improve the discussion. New ideas for discussion:
Central Dopamine System and Stroke (PMID: 30034335)
a. Stroke triggers early and massive dopamine release into the striatum.
b. Decrease in DA receptors following stroke.
c. Stroke can impact the response of the dopaminergic system to DA-modulating drugs
Other: The reviewer would appreciate it if the authors could provide the manuscript in the word format provided in the instruction for authors.
None
Author Response
Answers to Reviewer 4 Comments
Point 1. Abstract. Could the authors provide more data in the abstract? The data should be in numbers and percentages.
Answer 1: Thank you for your comments. I have added to and modified the parts you mentioned.
Our findings revealed a significant difference in the mean values of fractional anisotropy (FA) and tract volume (TV) of the MCT and MLT pathways between the patient and control groups (p<0.05). Specifically, the mean FA of the MCT and MLT showed a decrease of 7.94% and 6.33%, respectively, in the affected side compared to the control group (p<0.05). Similarly, the mean TV of the MCT and MLT showed a decrease of 73.22% and 78.79%, respectively, in the affected side compared to the control group (p<0.05). These changes were significantly different from those of the unaffected MCT, MLT, and control groups (p<0.05).
Point 2. Please limitations should be described in the last paragraph of the discussion.
Answer 2: Thank you for your comments. I have added to and modified the parts you mentioned.
This study had several limitations. First, DTT analysis generated false-positive and false-negative findings owing to the crossing fibers and partial volume effect, respectively. Second, fiber crossing and complexity may deem the accurate DTT depiction of the underlying fiber architecture difficult. Consequently, the DTT findings may have underestimated the significance of fibers in the neural tract. In addition to the limitations related to DTT analysis and lack of clinical correlation, the study was further constrained by a small sample size and lack of consideration for variability in age and stroke type among participants, which may have influenced the findings and their generalizability. Future research should overcome these limitations and conduct neuroimaging comparisons using clinical data.
Point 3. It is advised to improve the discussion. New ideas for discussion:
Central Dopamine System and Stroke (PMID: 30034335)
- Stroke triggers early and massive dopamine release into the striatum.
- Decrease in DA receptors following stroke.
- Stroke can impact the response of the dopaminergic system to DA-modulating drugs
Answer 3: Thank you for your comments. I have added to and modified the parts you mentioned.
Additionally, it is important to consider the intersection of the central dopamine system and stroke in the context of our findings. Stroke has been shown to trigger an early and massive release of dopamine into the striatum [25,26]. This could potentially exacerbate the microstructural damage in the MCT and MLT pathways observed in our study, as the sudden influx of dopamine might further disrupt the normal functioning of these pathways. Moreover, a decrease in dopamine receptors following stroke has been reported [26-28]. This decrease could potentially contribute to the observed lower mean values of FA and TV on the affected side of the MCT and MLT pathways, as the reduced number of dopamine receptors might impair the transmission of signals along these pathways. In the context of our study, this could mean that the observed damage to the MCT and MLT pathways might also affect the patient's response to treatment. These findings from the literature further underscore the severe impact of stroke, particularly ischemic stroke, on the brain's structure and function.
- Busto, R.; Dietrich, W. D.; Globus, M. Y. T.; Valdés, I.; Scheinberg, P.; Ginsberg, M. D. Small differences in intraischemic brain temperature critically determine the extent of ischemic neuronal injury. J. Cereb. Blood Flow Metab. 1987 7(6), 729-738.
- Gower, A.; Tiberi, M. The intersection of central dopamine system and stroke: potential avenues aiming at enhancement of motor recovery. Front. Synaptic Neurosci. 2018 10, 18.
- Unger, E. L.; Wiesinger, J. A.; Hao, L.;Beard, J. L. (2008). Dopamine D2 receptor expression is altered by changes in cellular iron levels in PC12 cells and rat brain tissue. J. Nutr. 2008, 138(12), 2487-2494.
- Bahk, J. Y.; Li, S.; Park, M. S.; Kim, M. O. Dopamine D1 and D2 receptor mRNA up-regulation in the caudate–putamen and nucleus accumbens of rat brains by smoking. Prog. Neuropsychopharmacol. Biol. Psychiatry 2002 26(6), 1095-1104.
